# Polymyxin B Peptide Hydrogel Coating: A Novel Approach to Prevent Ventilator-Associated Pneumonia

**DOI:** 10.3390/ijms251910269

**Published:** 2024-09-24

**Authors:** Milan Wouters, Laurence Van Moll, Linda De Vooght, Emilia Choińska, Joanna Idaszek, Karol Szlązak, Marcin K. Heljak, Wojciech Święszkowski, Paul Cos

**Affiliations:** 1Laboratory of Microbiology, Parasitology and Hygiene (LMPH), Faculty of Pharmaceutical, Biomedical and Veterinary Sciences, University of Antwerp, Wilrijk, 2000 Antwerp, Belgium; milan.wouters@uantwerpen.be (M.W.); laurence.vanmoll@uantwerpen.be (L.V.M.); linda.devooght@uantwerpen.be (L.D.V.); 2Faculty of Materials Sciences and Engineering, Warsaw University of Technology, Woloska 141, 02-507 Warsaw, Poland; emilia.choinska@pw.edu.pl (E.C.); joanna.idaszek@pw.edu.pl (J.I.); karol.szlazak@pw.edu.pl (K.S.); marcin.heljak@pw.edu.pl (M.K.H.)

**Keywords:** antimicrobial peptides, *Pseudomonas aeruginosa*, ventilator-associated pneumonia, polymyxin B, antibacterial coating, hydrogel coating

## Abstract

Ventilator-associated pneumonia (VAP) remains one of the most common hospital-acquired infections (HAI). Considering the complicated diagnosis and the lack of effective treatment, prophylactic measures are suggested as the new standard to prevent the disease. Although VAP often manifests a polymicrobial nature, *Pseudomonas aeruginosa* remains one of the pathogens associated with the highest morbidity and mortality rates within these mechanically ventilated patients. In this paper, we report on the development of an antibacterial hydrogel coating using the polymyxin B (PMB) peptide to prevent bacterial adhesion to the polymeric substrate. We fully characterized the properties of the coating using atomic force microscopy (AFM), scanning electron microscopy (SEM), wettability analyses and Fourier-transform infrared (FTIR) and Raman spectroscopy. Furthermore, several biological assays confirmed the antibacterial and anti-biofilm effect of the tubing for at least 8 days against *P. aeruginosa*. On top of that, the produced coating is compliant with the requirements regarding cytocompatibility stated in the ISO (International Organization for Standardization) 10993 guidelines and an extended release of PMB over a period of at least 42 days was detected. In conclusion, this study serves as a foundation for peptide-releasing hydrogel formulas in the prevention of VAP.

## 1. Introduction

Nosocomial infections are a major hurdle in intensive care units (ICU) all over the world. Many of these infections are related to medical devices such as urinary catheters, intravascular catheters or medical implants. However, according to a global meta-analysis by Raoofi et al., pneumonia remains one of the most frequently occurring types of nosocomial infections [1]. A total of 96% of hospital-acquired pneumonia (HAP) episodes were associated with mechanical ventilation and therefore classified as ventilator-associated pneumonia (VAP) [2]. According to the latest reports of the European Centre for Disease Prevention and Control (ECDC), VAP impacts 5–40% of patients who undergo mechanical ventilation for over two days [3]. VAP can be categorized into early-onset VAP, which occurs within 96 h of mechanical ventilation, and late-onset VAP, which occurs after 96 h of mechanical ventilation. Late-onset VAP is usually caused by multidrug-resistant (MDR) pathogens, which increase its morbidity and mortality. The most common causative MDR pathogens include *Pseudomonas aeruginosa*, *Acinetobacter baumannii and Klebsiella pneumoniae* [4]. On top of that, the International Nosocomial Infection Control Consortium reports a mortality rate of 17.12% in ICU patients without nosocomial infections. It escalates, however, to 42.32% in patients with VAP [4,5].

The lack of a single set of reliable diagnostic criteria makes it challenging to accurately identify VAP patients. This uncertainty in diagnosis hinders the effective treatment of VAP, resulting in empirical antibiotic therapy, ultimately contributing to antimicrobial resistance (AMR) [6,7]. Moreover, VAP is also associated with a significant economic impact. Hospital expenses for VAP patients are significantly higher than for ventilated patients without VAP due the required antibiotic treatment regimens and the prolonged mechanical ventilation [8]. The abovementioned data clearly show the need for new anti-infective medical devices.

When discussing anti-infective medical devices, varying approaches to surface engineering have been studied. Where passive techniques focus on changing the topography and hydrophilicity of the surface, active approaches attempt to incorporate antimicrobial substances on the surface. These modifications may include metal-based coatings, biocides or other bio-inspired surfaces [9,10]. One option to create such an anti-infective medical device is the incorporation of antimicrobial peptides (AMPs).

During the last decade, AMP immobilization strategies have seen a tremendous increase in preventing bacterial colonization and biofilm formation on medical devices [11,12]. A plethora of immobilization strategies have been described, such as polymer brushes, chemical coupling, layer-by-layer assemblies, encapsulation in the matrix, etc.; all associated with their own benefits and drawbacks [11,13,14]. This research paper addresses the possibility of incorporating the polymyxin B peptide (PMB) in an acrylic-based hydrogel matrix as a first step, with the goal of extrapolation to other, novel AMPs in a later stage. The acrylic component, as the main polymer of the hydrogel matrix, was used with the purpose of achieving good compatibility with the AMPs and the surrounding tissues [15,16]. 

PMB is a cationic peptide and, where other antibiotics often encounter AMR, PMB often retains activity against these MDR organisms [17]. It specifically targets Gram-negative pathogens by destabilizing the lipopolysaccharides present in the outer cell membrane [17,18]. The therapeutic window of PMB is narrow and endotracheal intubation is a prolonged process of approximately seven days. Therefore, a slow-release coating on these endotracheal tubes (ETTs) is preferred to maintain activity and at the same time avoid toxicity [17,19,20]. 

Within this work, we thus developed for the first time a peptide-releasing coating of PMB in a matrix of acrylic acid (AA) and sulfobetaine methacrylate (DMAPS) on a polyvinyl chloride (PVC) substrate. Both monomers are polymerized via an ultraviolet (UV) photopolymerization process which facilitates hydrogel formation and therefore localized dose-specific drug delivery [11,21]. The chemical composition, mechanical data and visual interpretation of the hydrogel-coated tubes were determined using scanning electron microscopy (SEM), Fourier-transform infrared spectroscopy (FTIR), dynamic mechanical analysis (DMA), atomic force microscopy (AFM), water contact angle (WCA) and Raman spectroscopy. Furthermore, the anti-adhesion and anti-biofilm effects were determined through zone-of-inhibition (ZOI) and viable plate count. The release of PMB was determined through a micro bicinchoninic assay (Micro-BCA) and its biocompatibility via a cell-based cytotoxicity test. Although this manuscript addresses the incorporation of PMB, the foundations of this coating are meant to be multifunctional and therefore to be extrapolated to novel AMPs.

## 2. Results

### 2.1. Antibacterial Hydrogel Coating Preparation

Modified from Liu et al., a new hydrogel preparation method is described for the coating of PVC ETTs in Figure 1 [22]. After several thorough optimization cycles, a fully stable coating with optimal antibacterial properties could be applied on the PVC surface. Since the used tubing has a very small diameter, a needle was used as a support to prevent the deformation of the ETTs during the coating process. The small dimensions of the ETTs were essential with the aim of future murine experiments and comparing these in vitro data with future in vivo results.

To minimize excessive thickness of the coating, yet ensuring its antibacterial effectiveness, a hydrogel skin on the surface was formed via a process called UV photopolymerization. The goal of this hydrogel was not only to create a matrix for PMB incorporation, but also to covalently attach carboxylic acid groups to the PVC substrate. This process was initiated with benzophenone, a hydrophobic photoinitiator, which makes the surface more prone to interaction with the hydrogel monomers [23,24]. Afterwards, the tubing was UV irradiated (365 nm) within a solution of hydrogel monomers, DMAPS and AA, with an optimized concentration of 3% and 5%, respectively. The combination of these monomers can be supported by two different reasons. First of all, carboxylic acids groups are required in the next step when coupling an amine-containing peptide to a hydrogel layer. Therefore AA, with one free carboxylic residue and its small size, was the best option in this setting. Secondly, it is proven that zwitterionic compounds such as DMAPS have excellent anti-fouling and biocompatibility properties [25,26]. Therefore, the combination of both monomers were assumed to be ideal.

The last crucial step in the process involved PMB loading and was performed using N-(3-Dimethylaminopropyl)-N′-ethylcarbodiimide hydrochloride (EDC) and N-hydroxysuccinimide (NHS)-mediated conjugation. This technique is widely used for covalent linkage of an amine-containing peptide to a carboxylic acid residue [27,28,29]. ETTs coated with an unloaded hydrogel were denoted as DMAPS_AA_PVC and ETTs with a PMB containing hydrogel were denoted as DMAPS_AA_PMB_PVC.

### 2.2. In-Depth Spectral Analysis of Coated Endotracheal Tubing

To acquire more information about the molecular and chemical composition of the coated ETTs, FTIR and Raman spectroscopy were both used as complementary techniques. The FTIR spectra of coated and uncoated samples are shown in Figure 2. Several transmission minima can be found in all three samples; the peaks located around 2930 cm^−1^ belong to C-H stretching, minima at 1428 and 963 cm^−1^ are due to CH_2_ bending, 1240 cm^−1^ is attributable to C-H bending near a C-Cl bond, the peaks at 1114 cm^−1^ can be assigned to C-H in plane bending vibration and the minima at 610 cm^−1^ belong to C-Cl stretching [30,31]. The peak identified at approximately 1724 cm^−1^ clearly indicates the C=O stretching vibration of an ester functional group, which is intrinsically not seen in pure PVC. One possible interpretation of this peak is the presence of dioctyl phthalate (DOP) as a plasticizer to ensure the flexibility of the PVC tubing [32]. Another explanation could be the aging of the PVC, which undergoes dechlorination and subsequently oxidation in the presence of oxygen [33]. The peaks at 1660 cm^−1^ could only be seen in DMAPS_AA_PVC and DMAPS_AA_PMB_PVC tubing and not in uncoated PVC. This specific peak could be attributed to the carbonyl stretch in AA. The subtraction spectrum of DMAPS_AA_PVC and DMAPS_AA_PMB_PVC can be found in Appendix A.

Lastly, the spectrum of the PMB peptide has several characteristic peaks which are absent in the spectra of the ETTs. The peaks in the upper region of the IR spectrum (3258, 3058 and 2960 cm^−1^) represent O-H stretching vibrations. Subsequently, clear amide I and II bands can be distinguished at 1643 cm^−1^ and 1525 cm^−1^, respectively. The last peak at 1069 cm^−1^ can be attributed to C-O stretching vibrations from alcohols or from the peptide bonds [34].

Raman spectroscopy was used as a complementary technique to FTIR characterization. As in FTIR, there are some characteristic peaks for the bare PVC as shown in Figure 3. The peak at 835 cm^−1^ represents CH_2_ rocking vibrations and 1069 cm^−1^ corresponds to C-C stretching vibrations. Peaks in the 1170–1438 cm^−1^ range typically represent alkyl signals. The peak at 1170 cm^−1^ is due to CH_2_ twisting, 1310 cm^−1^ is attributable to CH bending and 1438 cm^−1^ represents CH_2_ scissoring [35]. The peaks at 1611 cm^−1^ and 1729 cm^−1^ represent C=C stretching of an aromatic ring and C=O stretching of an ester group [36], respectively. Both serve as proof of phthalate incorporation as a plasticizer in the PVC ETTs. When analysing DMAPS_AA_PVC and DMAPS_AA_PMB_PVC samples, characteristic peaks show up at 1002 cm^−1^ and 1659 cm^−1^. These peaks represent the sulfonate group present in DMAPS (1002 cm^−1^) and the C=O stretch of AA (1659 cm^−1^) [37]. It has to be noted that no extra peaks were detected in the DMAPS_AA_PMB_PVC ETTs compared to the ETTs without PMB. Comparing the reference spectrum of the PMB peptide to the obtained spectra from the samples, the lack of characteristic PMB peaks is partially explained. While most of the peaks are small and therefore overlap with the peaks of the PVC substrate, the peaks at 979 cm^−1^ and 1004 cm^−1^ are strong and not present in the DMAPS_AA_PMB_PVC sample.

The subtraction spectrum of DMAPS_AA_PVC and DMAPS_AA_PMB_PVC can be found in Appendix A. The individual and extended spectra can be consulted in Appendix A. The extended spectra included characteristic PVC peaks at 636 cm^−1^ and 698 cm^−1^ representing C-Cl bond stretching and at 2915 cm^−1^ for CH_2_ vibrations [38].

### 2.3. Antibacterial Hydrogel Coating Enhances Surface Hydrophilicity

Following the chemical composition alteration of the coated tubes, the water contact angle was measured, since the wettability of the surface can alter bacterial adhesion as well as cell proliferation. A static WCA from a frontal and lateral point of view was applied to determine the wettability of the surface. Figure 4a shows a violin plot of the repeated measurements of the lateral WCA. An increase in wettability can be observed after hydrogel formation and after PMB incorporation. From this lateral point of view, uncoated PVC exhibits the highest WCA of 93.9° ± 3.7°, hydrogel-coated ETTs manifest a lower WCA of 80.0° ± 4.1° and PMB loading on this matrix increased the wettability of the surface, with a WCA equaling 70.7° ± 5.9° (Figure 4b–d). The violin plot and visual observation of the frontal wettability are presented in Appendix A.

The results from this experiment illustrate the high wettability of hydrogel-coated ETTs. Both AA and DMAPS have several hydrophilic groups, facilitating rapid water absorption and thus reducing the WCA [39,40]. On top of that, PMB has hydrophilic moieties as well, which can decrease the WCA even further [41]. Due to the small diameter of the tubing, gravitational forces have a more pronounced impact on the droplet and therefore the standard deviation of frontal measurements show more variability compared to the lateral WCA.

### 2.4. The Applied Coating Can Withstand Physiological Pressures

To analyze the mechanical properties of the coated ETTs, a compression test was conducted. A static force was applied in function of the strain for the different tubes (Figure 5). During the initial phase of compression, the transversal stiffness of uncoated PVC samples was observed to be greater than that of hydrogel-coated samples. Consequently, applying a hydrogel coating to the surface made the coated ETTs more compliant under compression up to the specified value of strain. For coated ETTs, the point at which the slope of the force–strain curve matches that of the uncoated samples can be identified. This point likely indicates the breakage of the hydrogel coating and equals 0.45 N and 0.77 N for DMAPS_AA_PVC and DMAPS_AA_PMB_PVC, respectively. Considering the surface area of the compressed tubing, this corresponds to 0.12 N/mm^2^ and 0.20 N/mm^2^. Such pressures are necessary to fracture the coating and further compression will lead to deformation of the bare PVC. When looking at the physiological significance, little is known about the exact tracheal pressure. Clinical trials suggest actual tracheal pressures up to 10–15 cmH_2_O (0.00098–0.0015 N/mm^2^), which are considerably lower than the required pressure to collapse the hydrogel coating [42,43]. The pressure inside the ETT (referred to as cuff pressure) can vary depending on the sex of the patient, the age of the patient, the type of ETT used, etc. To reduce mucosal damage and aspiration, the pressure on this cuff is often maximized at 30 cmH_2_O (0.0029 N/mm^2^) and therefore will also not damage the hydrogel coating [44,45].

### 2.5. Coated Endotracheal Tubes Exhibit a Rougher and More Textured Surface

To finalize the characterization of the coated samples, the surface of the coated ETTs was studied in full detail with atomic force microscopy to elucidate its topography and microstructure, both on areas of 25 µm^2^. On the one hand, the 2D microstructure map provides crucial insights on the structure and spatial arrangements of the surface, as shown in Figure 6a–c. On the other hand, topography focuses on the roughness of the sample and its texture as shown in the 3D images in Figure 6d–f.

The 2D microstructure images in Figure 6a–c show that uncoated PVC exhibits a relatively smooth surface with little to no distinct features, indicating its natural polymeric nature. The microstructure in the DMAPS_AA_PVC sample, however, changes to more textured areas, as well as areas which retain the same level of microstructure. Lastly, DMAPS_AA_PMB_PVC samples show the most textured surface, which proves the roughening of the surface by the coating procedure. The alteration in microstructure covers the whole scanned area, which implies that PMB incorporation indeed alters the corresponding features, such as hydrophilicity and mechanical strength, as discussed earlier.

The analysis of the 3D topography maps in Figure 6d–f confirms the findings of the surface microstructure. Although the difference in measured height only amounts up to several dozens of nanometers, the uncoated PVC can be classified as the smoothest. DMAPS_AA_PVC and DMAPS_AA_PMB_PVC exhibit bigger height differences and a rougher surface can be observed.

### 2.6. Scanning Electron Microscopy Confirms Anti-Biofilm Activity and Shows the Formation of Blister-Like Structures

The surface morphology of the different tubes and the effect of bacterial colonization were visually studied using SEM. As shown in Figure 7a, uncoated PVC shows no visible signs of handling, although being included in the cleaning and sterilization process. The surface of the tubing is smooth and therefore suitable for future coating procedures or in vivo experiments. Applying the DMAPS_AA hydrogel on the tubing (DMAPS_AA_PVC) does not alter the surface significantly; it becomes slightly rougher but remains smooth in general (Figure 7b). When looking at the DMAPS_AA_PMB_PVC tubing, the coating appears to be organized in patches rather than homogeneously on the surface. Some areas remain smooth, while other areas generate blister-like structures as shown in Figure 7c,d. These blister-like structures can reach diameters of approximately 10–15 µm within the observed patches. After incubation with *P. aeruginosa* (strain PA01), a clear biofilm is formed on the DMAPS_AA_PVC tubing. Bacteria, often covered in a bacterial biofilm matrix, can be distinguished on the whole surface of the tubing. This bacterial presence is also detected near the blisters on the tubing. The number of collapsed blisters seems to increase whenever *P. aeruginosa* is present, therefore resulting in “crater-like” structures (Figure 7e–g). It is not clear what causes the formation of these crater-like structures and the cause of the collapse of these blister-like structures. After incorporating the PMB peptide, no biofilm growth can be observed after 72 h on the DMAPS_AA_PMB_PVC tubing (Figure 7h–j). The blister- and crater-like structures are still abundant; however, no live bacteria could be detected. The surface near the blisters contains a large number of debris, which can be assumed to be the remnants of the bacterial biofilm matrix. Although the produced coating is not completely uniformly distributed on the whole surface, it can be concluded that the DMAPS_AA_PMB_PVC tubing is effective in inhibiting *P. aeruginosa* biofilm formation.

### 2.7. Polymyxin B-Coated Tubing Meets Biocompatibility Standards

Besides the mechanical and visual characterization of the coated ETTs, the produced tubing should also meet the standards of biocompatibility. According to the ISO (International Organization for Standardization) standards for medical devices (ISO 10993), three main biocompatibility tests are advised when developing new medical devices: cytotoxicity, sensitization and irritation [46,47]. While the last two are important in the later stages of device development, cytotoxicity is a prerequisite for further biological evaluation [48]. The cytotoxic effect of the tubing was determined using MRC5-SV2 lung fibroblasts, since they are lung-specific [49]. As the ETTs will be in direct contact with the mucosal membranes in vivo, a direct cytotoxicity assay was conducted. Hereby, the tubing covered 10% of the area of the well as required by the ISO guidelines.

To quantify the viability of the cells, a colorimetric assay based on bioreduction of the compound “3-(4,5-dimethylthiazol-2-yl)-5-(3-carboxymethoxyphenyl)-2-(4-sulfophenyl)-2H-tetrazolium” (MTS) was conducted. As shown in Figure 8, the viability of the cells incubated with all three samples did not significantly differ from the negative control (only medium) after 24 h of exposure to the ETTs. The positive control containing 10% dimethyl sulfoxide (DMSO) achieved a cell viability of less than 0.2%, which is classified by the ISO 10993-5 as cytotoxic, since it is below the threshold of 70% [46,49]. Therefore, it can be concluded that neither the hydrogel-coated ETTs nor the peptide incorporated in the hydrogel were cytotoxic. This result could initially be classified as rather unexpected, since the toxicity of PMB is well-described in the literature [50,51]. Most of the published toxicity tests, however, are performed on cells derived from kidney-based tissues, such as the adrenal medulla, proximal tubules or epithelial tissues [52]. The cytotoxicity of the PMB on lung epithelial cells is less studied. It is shown that that a concentration of 1.74 mM (±1.30 mg/mL) is required to induce 50% of maximal cell death in lung epithelial A549 cells [53]. As shown below, the amount of PMB released from the coating is below this threshold and therefore not expected to be cytotoxic.

On top of that, after 24 h of co-incubation of the samples with the fibroblasts, normal cell morphology was retained across all groups and there was no change in cell confluency in the proximity of the tubing. These results are in line with the quantitative viability analysis. Visual analyses of the cells can be consulted in Appendix A.

### 2.8. Detection of the Extended Release of Polymyxin B from the Hydrogel Matrix

The amount of PMB released from the hydrogel coating over time was quantified as shown in Figure 9a. There are multiple ways of determining the concentration of PMB in solution, such as high-performance liquid chromatography, mass spectrometry, etc. In this experiment, the amount of polymyxin B released from the PVC ETTs was measured through a standardized micro-BCA assay [54,55]. Initially, a burst release of approximately 5 µg/mL PMB can be observed after 3 h of incubation in 500 µL Dulbecco’s phosphate buffered saline (DPBS). The release profile decreased and reached a minimum at 72 h. Afterwards, it stabilized and turned into a sustained release profile for the following 42 days. The burst release can probably be attributed to a higher amount of PMB being incorporated in the outer layer of the hydrogel coating. The observed release, however, is inconsistent, since the standard deviation varies from 1.5 µg/mL to 8.5 µg/mL. This fluctuation could be explained by variable concentrations of PMB in the outer layer of the hydrogel, which could be attributable to insufficient washing after coating the ETTs, thereby leaving unbound PMB on the surface. The sustained release profile remains constant at approximately 2.6 µg/mL, which is higher than the minimal inhibitory concentration (MIC). Therefore, the coating can be classified as antibacterial for up to at least 42 days. Figure 9b shows the cumulative release of PMB throughout the 42 days, where it reaches its observed maximum at 30 µg/mL.

### 2.9. Confirmation of the In Vitro Antibacterial Activity against Various P. aeruginosa Strains

Lastly, the produced tubing was subject to different antibacterial tests. Since *P. aeruginosa* is one of the most common and most deadly VAP-associated pathogens, three different strains of this specific bacterium are included [4]. PA01 is a common laboratory strain, which is fully sequenced and representative of moderately virulent *P. aeruginosa* phenotypes [56]. RP73 and AA44 are clinical isolates from cystic fibrosis patients with a chronic *P. aeruginosa* infection [57,58,59].

Firstly, as shown in Figure 10a, the ability of *P. aeruginosa* to attach to the foreign PVC substrate within a time period of 3 h was assessed. Strain PA01, AA44 and RP73 show a mean initial adhesion on uncoated PVC of 4.4 × 10^3^, 2.1 × 10^3^ and 2.4 × 10^2^ CFU/mm^2^, respectively. The initial adhesion on hydrogel-coated PVC (DMAPS_AA_PVC) follows the same trend with mean initial adhesions for PA01, AA44 and RP73 of 5.7 × 10^3^, 1.9 × 10^3^ and 1.1 × 10^2^ CFU/mm^2^, respectively. The initial bacterial adhesion on PMB-incorporated hydrogel tubing (DMAPS_AA_PMB_PVC) did not reach the quantification limit of 1.5 × 10 CFU/mm^2^. Afterwards, an anti-biofilm assay was performed as shown in Figure 10b, where the ability to prevent the formation of a mature biofilm after 72 h on different produced ETTs was analyzed. Uncoated PVC exhibits a mean adhesion of 1.8 × 10^4^, 3.2 × 10^4^ and 6.8 × 10^2^ CFU/mm^2^ for PA01, AA44 and RP73, respectively. Hydrogel-coated PVC shows a mean bacterial adhesion of 1.8 × 10^4^, 2.9 × 10^4^ and 1.8 × 10^4^ CFU/mm^2^ for PA01, AA44 and RP73, respectively. PMB-incorporated hydrogels showed no biofilm formation of PA01 and AA44; however, two outliers were detected for RP73, showing a bacterial adhesion of 1.1 × 10^2^ and 1.2 × 10^2^ CFU/mm^2^.

The antimicrobial activity of the tubing was also evaluated over time against *P. aeruginosa* PA01, AA44 and RP73 using a modified Kirby–Bauer assay for 8 days, as shown in Figure 10c,d. The uncoated and coated ETTs were placed upright in freshly inoculated agar to visually check the ability to inhibit bacterial growth on nutrient agar. The DMAPS_AA_PMB_PVC tubing showed a ZOI of 7.9, 7.5 and 8.7 mm for PA01, AA44, RP73, respectively, on the first day. A gradual, but slow, decrease in this zone was observed with a remaining ZOI of 5.6 mm (PA01), 3.6 mm (AA44) and 5.5 mm (RP73) on the 8th day. These results confirm the presence of antimicrobial activity of the tubes for at least 8 days. Both uncoated PVC as DMAPS_AA_PVC showed no ZOI.

## 3. Discussion

The aim of this research paper was to develop a novel drug-eluting antibacterial coating against *P. aeruginosa* in order to prevent cases of VAP. Although *P. aeruginosa* ranks lower on the WHO Bacterial Priority Pathogens List in 2024 compared to 2017, it still remains a critical pathogen, especially when focusing on VAP [60]. Not only are the significant numbers of this pathogen highlighted in a recent systematic review by Velásquez-Garcia et al. but the rising resistance must also be taken into account [60]. Although numbers are higher in cases concerning *K. pneumoniae*, carbapenem resistance and extended-spectrum beta-lactamase resistance patterns are still detected for *P. aeruginosa* [61,62].

A novel antibacterial medical device which facilitates the prevention of VAP could reduce its incidence and consequently the problems related to AMR. As described by Marcut et al., several implantable drug-releasing ETTs are under investigation; however, only one has made it through the full commercialization process to date [9]. It involves a silver nanoparticle coating, which is well-described in the setting of VAP, but not encouraged due to the discrepancy in toxicity and effectiveness data [9,10]. For this reason, an all-encompassing approach involving novel antibacterial compounds, such as AMPs, is much-needed [63]. This paper tries to serve as a cornerstone in this evolving field of antibacterial coatings by designing a polymeric coating consisting of PMB via a UV photopolymerization process.

In the first crucial step, a comprehensive characterization of coated ETTs offers significant insights into the coating’s mechanical properties, chemical composition and microstructure. First, the outcomes of the mechanical tests conclude that the coated ETTs are more compliant under pressure and that the novel drug-eluting antibacterial coating can withstand the physiological loads characteristic for ETTs. Second, applying the hydrogel coating on the surface significantly increases its wettability. This wettability even further increases when the peptide is incorporated in the hydrogel. It is reported that bacterial adhesion is not promoted in extreme conditions such as a superhydrophilic (WCA~0°) or superhydrophobic (WCA > 150°) substrates [64,65]. Since the WCA of all the samples was found in a more moderate region (WCA~90°), bacterial adhesion could occur on both the uncoated PVC and DMAPS_AA_PVC [64,66].

Bacterial adhesion is not only altered by the wettability of the surface, its topography, roughness and elemental composition play a role in bacterial attachment to the surface as well. Both SEM and AFM prove the roughening of the surface and a textured microstructure in the DMAPS_AA_PVC ETTs. The presence of blister-like structures on the surface does not hinder the formation of a bacterial biofilm, as a clear bacterial network is observed after 72 h. The similar pattern of bacterial presence on the surface detected after both 3 h and 72 h, can be explained by the increased roughness of the coating which is outbalanced by an increased wettability. Initially, a decrease in bacterial attachment was expected, due to the zwitterionic nature of DMAPS and its anti-fouling properties [67]. It is possible that there was an inability of the monomer to crosslink within the hydrogel and form a stable polymeric network. It is proven that longer DMAPS segments have greater antibacterial effects [67]. Although no antibacterial properties of DMAPS could be detected, the presence of its characteristic sulphonate group was detected via Raman spectroscopy. On top of that, the ester group of AA was detected in both Raman spectroscopy and FTIR.

When PMB is incorporated, no bacterial growth is detected on SEM images and the AFM indicates an even more textured microstructure and thus a rougher surface. It should, however, be noted that not all parts of the ETTs seem homogenously coated. Specific regions contain a rough surface with protrusions, while other parts only display a minimal change compared to uncoated PVC. Although a rougher surface results in more bacterial adhesion, no bacterial presence can be detected in the DMAPS_AA_PMB_PVC tubing, assumingly due to the presence of PMB [68,69]. Its presence, however, could not be detected via either FTIR or Raman spectroscopy. Most of the characteristic peaks in Raman spectroscopy are overlapped by PVC signals and the characteristic peaks at 979 cm^−1^ and 1004 cm^−1^ could be explained by the nature of the PMB peptide, which is a sulphate salt. These peaks are therefore generated by sulphate and sulphonate functional groups, which are lost during the rinsing step in the coating process [70,71,72]. On top of that, the PMB peptide does generate some specific peaks in the FTIR spectrum that were not detected in the DMAPS_AA_PMB_PVC spectrum. The absence of these characteristic absorption bands is hypothesized to result from the low concentration of PMB incorporated in the hydrogel matrix, which could not be detected due to the detection limitations of the attenuated total reflectance (ATR) infrared spectroscopy. ATR-FTIR is generally limited to a penetration depth of only a few micrometers. Only a minimal amount of PMB is incorporated in the hydrogel and the potential matrix interferences from the hydrogel likely contributed to the non-detection of PMB-specific peaks in the spectrum [73,74,75].

Subsequently, despite the fact that the PMB peptide is often related to acute toxicity, the produced ETTs exhibit an excellent biocompatibility [76]. One explanation for this good outcome is the covalent immobilization of the peptide, because often, AMP-based delivery systems are associated with a significant toxicity. It, however, decreases when peptides are covalently bonded within a hydrogel network [76,77]. On top of that, both of the hydrogel components (DMAPS and AA) did not show any signs of cellular toxicity when incorporated in the coating.

Although PMB is partially covalently linked within the hydrogel to the AA moieties, a release of the peptide can be detected via the micro-BCA assay. It is assumed that the peptide therefore is coated in two different manners to the PVC ETTs. Firstly, a fraction of the PMB peptide is covalently bound to the PVC substrate via the AA functional groups, which serve as a linking layer [11]. These attached peptides ensure a so-called kill-on-contact layer, which inhibits the adhesion of bacteria to the substrate [78]. Secondly, since the produced hydrogel is treated with an excess of the PMB peptide, it can be assumed that a part of it will be incorporated in the matrix structure and thus, can be released over time [12,27]. When discussing such an extended-release of drugs out of a polymeric matrix, a burst release is often described, before following first- or second-order kinetics [79,80]. Although the obtained standard deviation is variable, our hypothesis remains that the PMB accumulated in the outer layer is released in a few hours as a burst profile. No definite conclusion can be made about the absolute duration of drug release from the hydrogel; however, it can be classified as antibacterial for up to 42 days. This is definitely lengthy enough to prevent VAP, since approximately 90% of mechanically ventilated patients are liberated from the ventilator within 14 days [81]. Lastly, when looking at the obtained antibacterial results, the capability of the DMAPS_AA_PMB_PVC tubing to completely prevent bacterial adhesion after 3 h is a crucial part in its preventive goals. Colonization of the ETTs happens in almost every patient during handling or insertion through the upper airway [82]. Another noticeable difference is the lower adhesion of strain RP73 to the uncoated PVC as well as the DMAPS_AA_PVC. This is probably due to the intrinsic properties of the strain, specifically the low-inflammatory lipo-oligosaccharide molecules that possess under-acylated lipid A structures, which are partly responsible for bacterial adhesion to the surface [83,84]. The observed differences between RP73 and the other strains in the biofilm formation are less pronounced, since the bacteria have 72 h to attach and thereby reach their stationary phase.

In conclusion, a streamlined all-encompassing coating strategy of ETTs involving novel AMPs is lacking. AMPs are a viable and sustainable alternative and possess the required characteristics to be incorporated in novel antimicrobial coatings. When developing a coating on ETTs, some crucial requirements have to be met like drug-eluting over a long period of time, biocompatibility and antibacterial and anti-biofilm activities. Although the produced coating fulfils these conditions, there is always room for improvement; namely, the uniformity of the coating should be improved (by further optimizing the coating process with, e.g., shaking steps) and the final amount of PMB incorporated should be determined. Although this manuscript solely focuses on PMB incorporation, it is meant as a versatile foundation for AMP-based surface modifications to prevent VAP. Within this work, considerable progress has been made, but optimization and in vivo studies are recommended to fully realize the clinical potential of this antibacterial coating technology.

## 4. Materials and Methods

### 4.1. Materials and Biologicals

PVC tubing (Tygon^®^ ND-100-80) was bought from Qosina Corp. (Ronkonkoma, NY, USA). MRC5-SV2 lung fibroblasts, sodium dodecyl sulfate (SDS), benzophenone, [2-(Methacryloyloxy)ethyl]dimethyl-(3-sulfopropyl) ammonium hydroxide (DMAPS), anhydrous acrylic acid (AA), N-(3-Dimethylaminopropyl)-N′-ethylcarbodiimide hydrochloride (EDC), N-hydroxysuccinimide (NHS), MES hydrate, isopropanol, glutaraldehyde, sodium cacodylate buffer, saccharose, hexamethyldisilizane, Luria Bertani (LB) agar and broth were all purchased from Merck (Darmstadt, Germany). Absolute ethanol, polymyxin B(PMB, purity ≥ 80%), Dulbecco’s Modified Eagle Medium (DMEM), DPBS, calcium chloride dihydrate and tris(hydroxymethyl)aminomethane (TRIS) buffer were bought from Thermo Fisher Scientific (Waltham, MA, USA). MTS was purchased from Promega Corporation (Madison, WI, USA). *P. aeruginosa* ATCC 1596 (PA01) was obtained from the ATCC (American Type Culture Collection, Manassas, VA, USA) and strain RP73 and AA44 were gifted.

### 4.2. Preparation of PVC Tubing

The tubing was prepared from commercially available Tygon^®^ ND-100-80 tubing with an inner and outer diameter of 0.25 mm and 0.76 mm, respectively. The coil was cut into 5 mm fragments using a 3D-printed model. The small fragments were first cleaned using a 1% SDS solution, followed by 10 min ultrasonic cleaning (Branson 2510E-MT, Branson, Brookfield, CT, USA). Subsequently, the tubing was subject to sterilization with absolute ethanol in a 15 min sonication step. Finally, the tubing was sonicated in sterile double-distilled water (ddH_2_O) for 10 min and air-dried before starting the experiments listed below.

### 4.3. Hydrogel Coating Application

Derived and modified from Liu et al., the production of the hydrogel coating consisted of placing the sterilized PVC fragments onto BD^®^ MicrolanceTM 30G needles in order to avoid deformations later in the process [22]. The mounted tubing was then submerged in a 10 wt.% benzophenone in isopropanol solution for 5 min, followed by 7 min of air-drying. Using sterile and sealed containers, the tubes were then immersed in DMAPS:AA (3 wt.%:5 wt.%) in sterile ddH_2_O and simultaneously UV-irradiated for 15 min (365 nm) with a distance of 5 cm between the sample and the UV source (9 W). Post-irradiation, the tubing was immersed in sterile ddH_2_O for 24 h to remove residual monomers. During the next step, the hydrogel coating was activated using EDC (0.32 M) and NHS (0.10 M) in MES buffer (pH 5.5). The activated tubing was washed twice with ddH_2_O and treated with 1 mg/mL PMB for 24 h before being vacuum dried. Finally, three groups of the ETTs were prepared: uncoated (uncoated PVC), PVC coated with the DMAPS_AA hydrogel (DMAPS_AA_PVC) and hydrogel-coated PVC with PMB incorporation (DMAPS_AA_PMB_PVC).

### 4.4. Fourier-Transform Infrared Spectroscopy and Raman Spectroscopy

The characteristics of the coated ETTs were analyzed using ATR-FTIR using a Bruker alpha II spectrophotometer. Each analysis consisted of 32 scans with a resolution of 4 cm^−1^ and the data were gathered in a range from 4000 cm^−1^ to 600 cm^−1^. Raman measurements were conducted on an inVia Raman microscope (Renishaw, Wotton-under-Edge, UK). A diode laser with a 785 nm wavelength and 10% of power, 1200 L/mm grating and a 50× objective were used. The spectra in the extended range (Raman shift of 400–3300 cm^−1^) were registered with an accumulation of 2 and exposure time of 10 s. The spectra in the narrow range (Raman shift 788–1857 of cm^−1^) were registered in static mode, with an accumulation of 2 centered at 1350 cm^−1^ and exposure time of 10 s.

### 4.5. Wettability (Water Contact Angle)

To determine the surface wettability of the coated ETTs, the water contact angle was determined using the sessile drop method by means of a goniometer (Contact Angle System, OCA, Dataphysics). ETTs of each group (uncoated, DMAPS_AA_PVC and DMAPS_AA_PMB_PVC) were visualized in the longitudinal and frontal views and the water drop size equaled 0.3 µL and 0.1 µL, respectively. The measurements of wettability were performed at room temperature and snapshots were taken 2 s after applying the droplet.

### 4.6. Compression Mechanical Analysis

Samples were incubated in DPBS at 37 °C for 20 h before mechanical analysis and were tested in the wet state to prevent them from excessive drying [85]. A Q800 (TA Instruments, New Castle, DE, USA) instrument equipped with compression clamps was used for the analysis. Measurements were carried out at room temperature (22 °C), using a strain rate of 5%/min. The strain (%) was measured as a function of static force applied (N). During the test, all the tubular samples were subjected to lateral compression. The hydrogel breaking point was identified with a slope analysis of the derivative function in Python 3.12, with a tolerance level of 0.01, defined as:(1)$$\left|\frac{dy1}{dx}-\frac{dy2}{dx}<tolerance\right|$$
where *dy*1 is the first derivative of the strain–force data of uncoated PVC, *dy*2 is the first derivative of the strain–force data of DMAPS_AA_PV or DMAPS_AA_PMB_PVC and *dx* is the change in the applied force. The software and computer code used for the analysis are presented in Appendix A.

### 4.7. Atomic Force Microscopy

Microstructure and topography visualization of coated and uncoated ETTs was performed using an MFP 3D BIO microscope (Asylum Research/Oxford Instruments, Santa Barbara, CA, USA). Stable and controllable experiment conditions were ensured by closing AFM in a custom-made chamber that precludes temperature fluctuations. The microscope was operating in tapping mode. To register high-resolution topographical maps, a TAP190DLC (Budget Sensors, Watsonville, CA, USA) scanning probe was used. According to the producer, the probe is characterized with a spring constant of approximately 48 N/m. Samples were imaged over an area of 5 × 5 μm at 0.6 Hz. IgorPro (version 6.17), a professional, dedicated software provided by the microscope’s producer, was used for data analysis.

### 4.8. Scanning Electron Microscopy

The tubes were immersion-fixed overnight in 2.5% glutaraldehyde in 0.1 M sodium cacodylate buffer (supplemented with 0.05% CaCl_2_·2H_2_O; pH 7.4) in a shaker incubator at room temperature at 100 RPM. After fixation, the tubes were rinsed three times for 20 min each with 0.1 M cacodylate buffer (containing 7.5% saccharose). Subsequently, the tubes were dehydrated in an ascending series of ethanol (50, 70, 90 and 95%, 10–15 min each, and 100%, three times 30 min each), followed by overnight incubation in hexamethyldisilane at room temperature. Following another 4 h of air-drying, the tubes were mounted on a stub with carbon tape and gold-coated using a Sputter Coater. Samples were visualized in a JSM-IT100 (Jeol, Tokyo, Japan).

### 4.9. Cytotoxicity Evaluation

The cytotoxicity of the coating on the ETTs was tested by using MRC5-SV2 embryonic lung fibroblasts. Cells were cultured in DMEM (supplemented with 5% heat-inactivated fetal bovine serum) before the experiment. A 96-well plate (TPP, Trasadingen, Switzerland) was seeded with 100 µL cell suspension in each well (10,000 cells/well) and the fibroblasts were grown for 24 h until 80% confluency (37 °C, 5% CO_2_). Subsequently, the tubing was placed in quintuple in the middle of each well for each of the three groups. After 24 h of incubation (37 °C, 5% CO_2_), the samples were imaged using a Zeiss Primovert microscope (Carl Zeiss, Oberkochen, Germany). To evaluate the potential cytotoxicity quantitatively, 100 µL of fresh medium without fetal bovine serum and 20 µL of MTS reagent were added to each well and incubated at cell culture conditions for 1 h. To stop the reaction, 25 µL of 10% SDS solution was added and the absorbance was afterwards measured at 495 nm by means of a plate reader (FLUOstar Omega, BMG Labtech, Germany). The negative control was cells cultured in the wells which did not contain ETTs, and the positive control consisted of culture medium supplemented with 10% (*v*/*v*) of DMSO.

### 4.10. Polymyxin B Release Quantification

DMAPS_AA_PMB_PVC and DMAPS_AA_PVC tubes were immersed in 0.5 mL of DBPS and incubated at 37 °C with constant shaking (120 RPM/min) for different durations (3, 6, 16, 24, 48 and 72 h and, additionally, 7, 14, 21, 28, 35 and 42 days). A total of 150 µL was collected in triplicate at each time point and the amount of PMB in the supernatant was analyzed with a micro-BCA assay. A total of 150 µL of the sample was added to 150 µL of BCA working reagent in a microplate (Greiner, Germany), mixed on a plate shaker for 30 s and incubated for 2 h at 37 °C. Subsequently, the absorbance was measured at 562 nm with a microplate spectrophotometer (VANTAstar, BMG LABTECH, Ortenberg, Germany). To validate the process, pre-made stock solutions with a known concentration of PMB were plotted against the standard curve to verify the accuracy and correctness of the used method for PMB. The DMAPS_AA_PMB_PVC results were compared and corrected against the DMAPS_AA_PVC outcomes.

### 4.11. Antibacterial Evaluation

The antibacterial capacity of the coated tubing was tested by placing it in a bacterial suspension of a specific *P. aeruginosa* strain (PA01, AA44 or RP73) with an optical density (OD) of 0.02, measured and grown in LB broth. After 3 or 72 h, for the anti-adhesion or anti-biofilm assay, respectively, the bacterial suspension was removed, the tubes were washed twice with DPBS and transferred to a 0.05% Tween 80 in LB broth solution. This was followed by vortexing (20 s), 15 min sonication and another vortex cycle (20 s). Subsequently, the obtained solutions were diluted and plated onto LB agar for viable plate count analysis.

### 4.12. Statistical Analysis

Statistical analyses were conducted on data transformed to the log10 scale using the software GraphPad Prism v10.2.3. The Shapiro–Wilk test determined data normality. When data followed a normal distribution, a one-way analysis of variance (ANOVA) or unpaired *t* test was employed for assessing significant differences. For data that did not pass the Shapiro–Wilk test, the Kruskal–Wallis or Mann–Whitney *U* test was applied instead. When different strains are involved in the same grouped experiment, multiple unpaired *t* tests combined with the false discovery rate for multiple comparisons determined the significance level of the data.

## Figures and Tables

**Figure 1 ijms-25-10269-f001:**
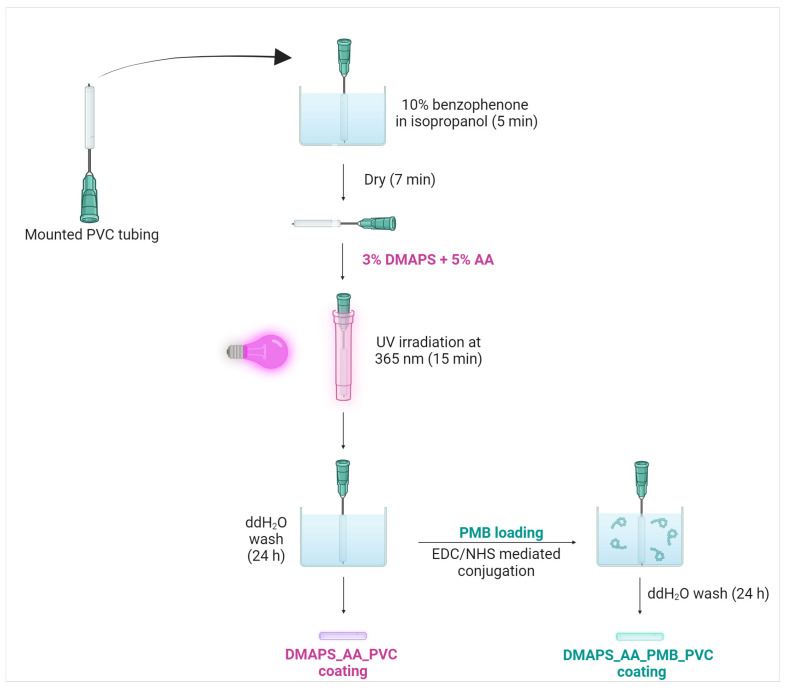
Protocol of hydrogel application. Polyvinylchloride (PVC) tubes were mounted on a needle before activation of the surface. The tubing was UV irradiated when submerged in sulbobetaine methacrylte (DMAPS) and acrylic acid (AA). After washing in double-distilled water (ddH_2_O), the tubing was activated using N-(3-Dimethylaminopropyl)-N′-ethylcarbodiimide hydrochloride (EDC) and N-hydroxysuccinimide (NHS), before loading of polymyxin B (PMB).

**Figure 2 ijms-25-10269-f002:**
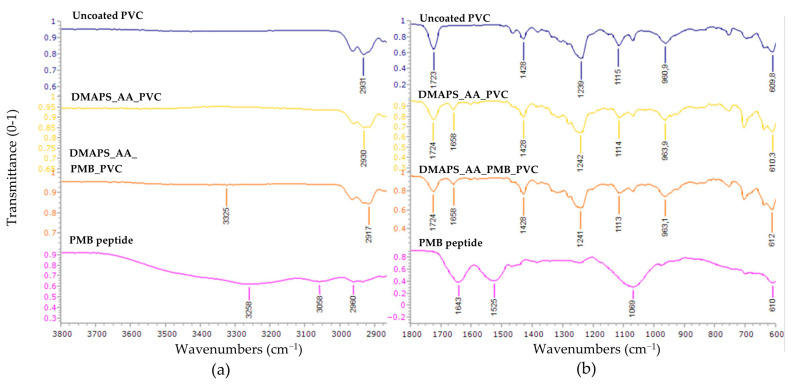
Fourier-transform infrared spectroscopy (FTIR) analysis. Subplot (**a**) represents the 3800 cm^−1^–2900 cm^−1^ region and (**b**) represents the 1800 cm^−1^–600 cm^−1^ region. No significant peaks were observed in the 2825 cm^−1^–1800 cm^−1^ region. The upper graph represents uncoated polyvinylchloride (PVC), the middle ones hydrogel-coated PVC (DMAPS_AA_PVC) and polymyxin B (PMB)-loaded hydrogel on PVC (DMAPS_AA_PMB_PVC), while the bottom graph represents the PMB peptide.

**Figure 3 ijms-25-10269-f003:**
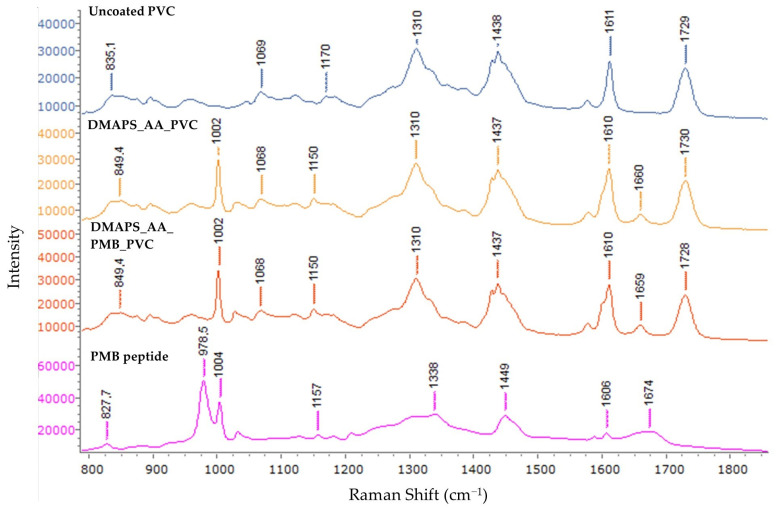
Raman spectroscopy. The upper graph represents uncoated polyvinylchloride (PVC), the middle ones hydrogel-coated PVC (DMAPS_AA_PVC) and polymyxin B (PMB)-loaded hydrogel on PVC (DMAPS_AA_PMB_PVC) and the bottom one the reference spectrum of PMB.

**Figure 4 ijms-25-10269-f004:**
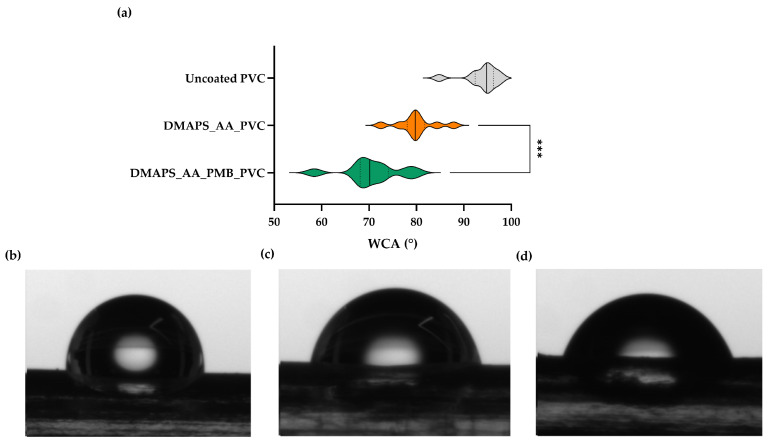
Lateral water contact angle. (**a**) Violin plot of the water contact angle (WCA) calculated as the median (full vertical line) of 10 measurements ± the standard deviation. The dotted lines represent the first and third quartile. An unpaired *t* test with Welch correction was used to test for statistical difference (*** *p* ≤ 0.001). (**b**–**d**) Water contact angle image of (**b**) uncoated PVC, (**c**) DMAPS_AA_PVC and (**d**) DMAPS_AA_PMB_PVC.

**Figure 5 ijms-25-10269-f005:**
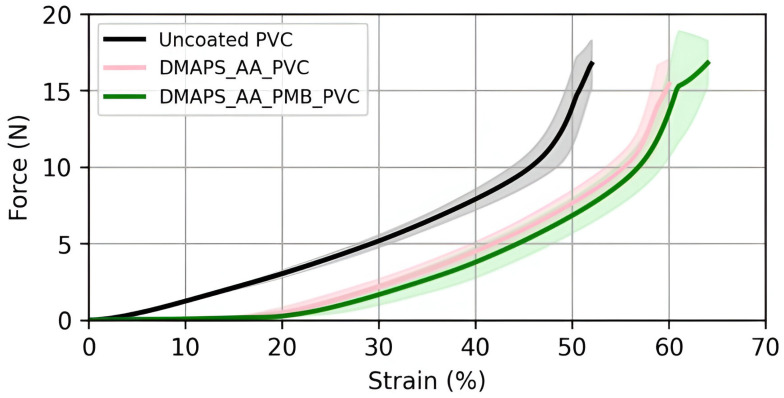
Mechanical analysis. Mechanical properties of uncoated PVC, DMAPS_AA_PVC and DMAPS_AA_PMB_PVC samples subjected to lateral compression.

**Figure 6 ijms-25-10269-f006:**
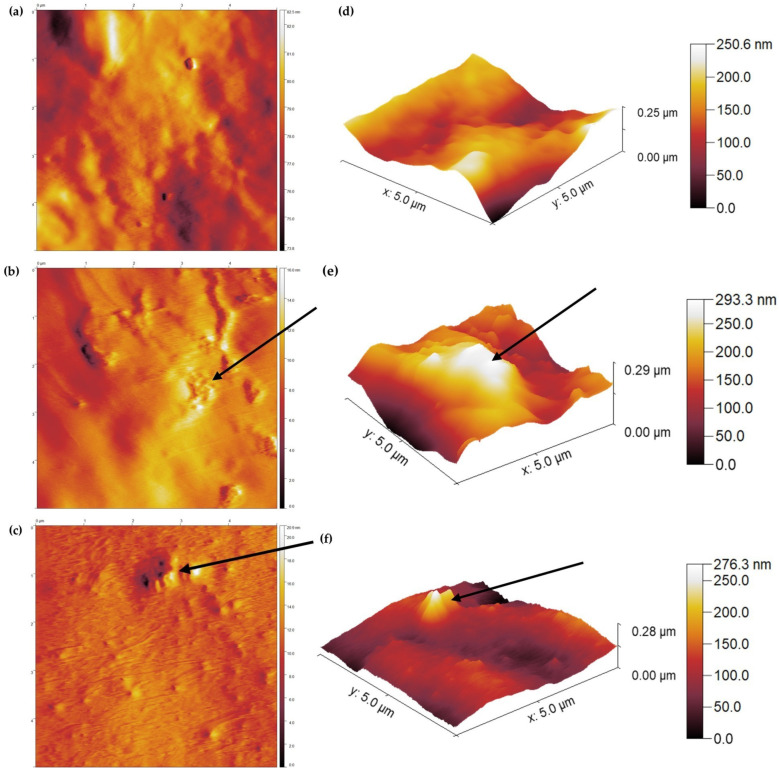
Atomic force microscopy. Microstructure (**a**–**c**) and topography (**d**–**f**) images of uncoated PVC (**a**,**d**), DMAPS_AA_PVC (**b**,**e**) and DMAPS_AA_PMB_PVC (**c**,**f**). Arrows indicate corresponding areas in microstructure and topography images.

**Figure 7 ijms-25-10269-f007:**
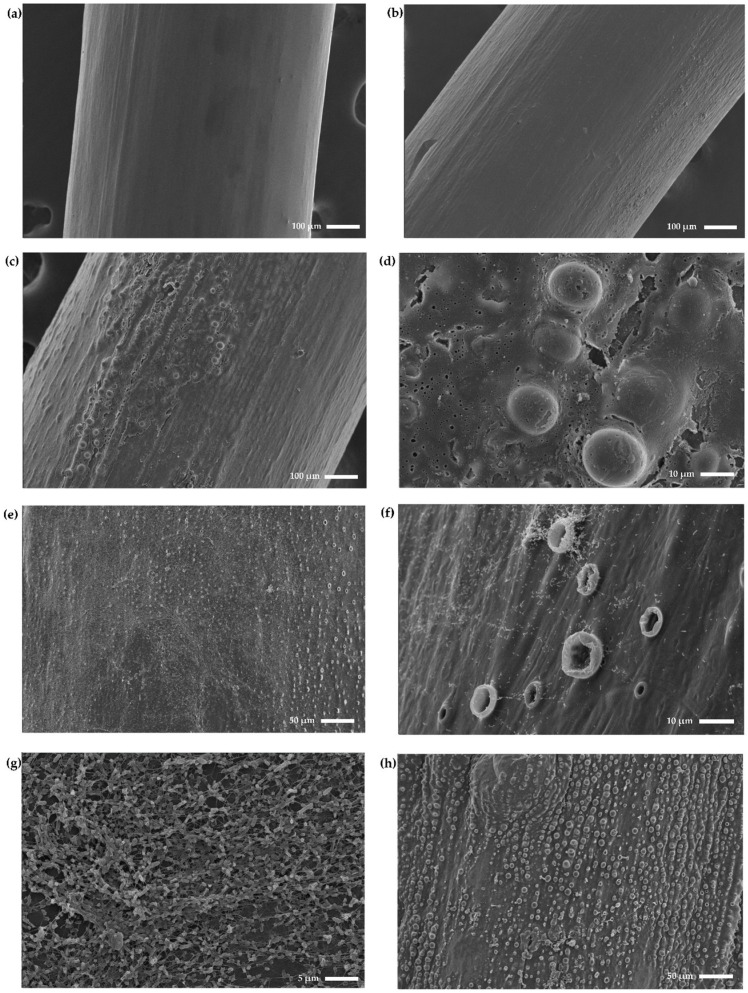
Scanning electron microscopy images. (**a**) Uncoated polyvinyl chloride (PVC), (**b**) DMAPS_AA_PVC, (**c**,**d**) DMAPS_AA_PMB_PVC, (**e**–**g**) PA01 biofilm growth (72 h) on DMAPS_AA_PVC and (**h**–**j**) PA01 biofilm growth (72 h) on DMAPS_AA_PMB_PVC.

**Figure 8 ijms-25-10269-f008:**
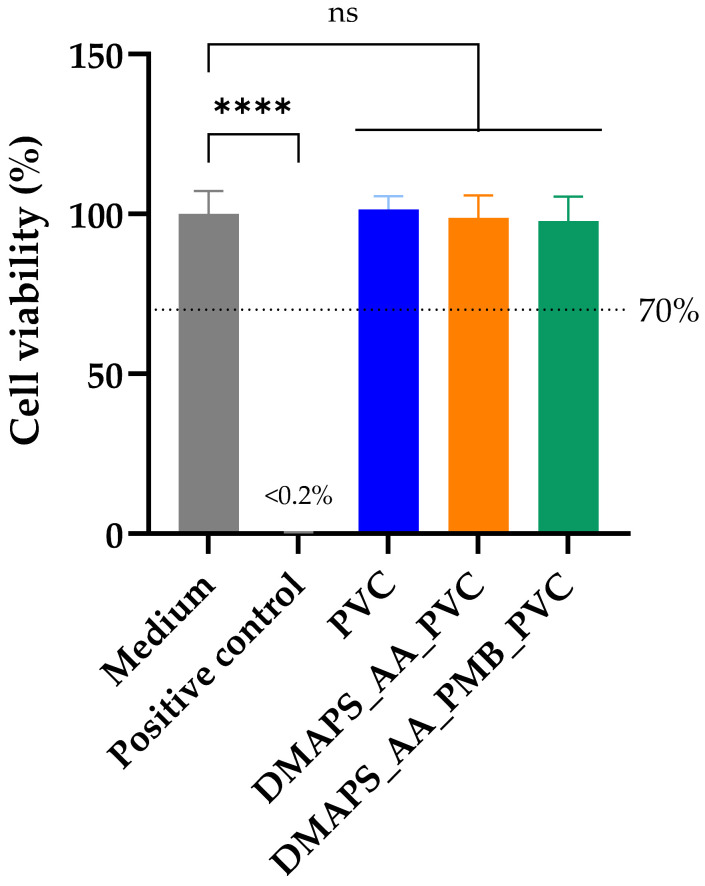
Cytotoxicity evaluation. Viability of MRC5-SV2 cells after 24 h of exposure to the endotracheal tubing. Normal medium was used as negative control and 10% dimethyl sulfoxide (DMSO) as a positive control. The level of significance was determined via an ordinary one-way ANOVA (*n* = 5; ns, not significant, *p* > 0.05 and **** *p* ≤ 0.0001) and a cell viability below 70% was considered cytotoxic.

**Figure 9 ijms-25-10269-f009:**
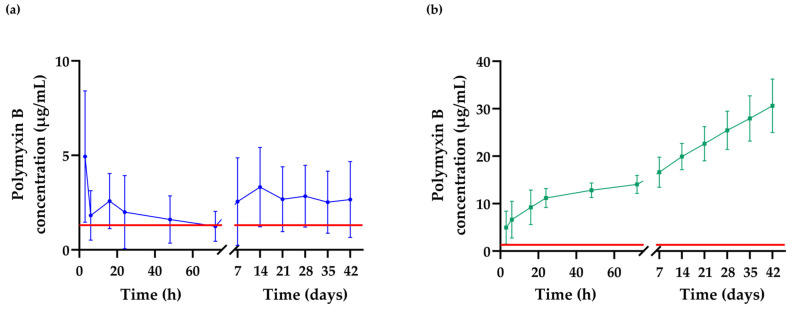
Polymyxin B release quantification. (**a**) Periodic and (**b**) cumulative release of PMB from DMAPS_AA_PMB_PVC tubing determined via a micro bicinchoninic assay (micro-BCA). The y-axis represents the amount of polymyxin B released per sample in 500 µL. The horizontal line at 1.3 µg/mL (1 µM) represents the minimal inhibitory concentration (MIC) of PMB. The release assay was performed in triplicate in two independent repeats, *n* = 6.

**Figure 10 ijms-25-10269-f010:**
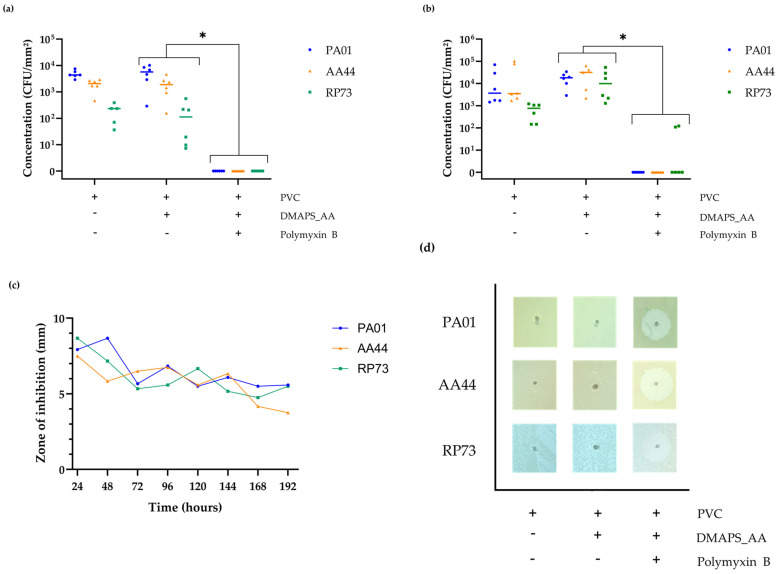
Antibacterial evaluations. (**a**) Inhibition of initial adhesion of different *P. aeruginosa* strains on uncoated polyvinyl chloride (PVC), hydrogel-coated PVC (DMAPS_AA_PVC) and polymyxin B (PMB)-loaded hydrogel on PVC (DMAPS_AA_PMB_PVC). (**b**) Inhibition of biofilm formation of different *P. aeruginosa* strains in the same conditions as described before. Both experiments were performed in biological duplicate with three technical measurements each (*n* = 6). The mean and individual data points are shown. Unpaired *t* tests with Welch correction and multiple testing correction via false discovery rates were used to test for statistical differences (** p* ≤ 0.05 (**c**,**d**) Quantitative and visual zone-of-inhibition (ZOI) assays determined via a modified Kirby–Bauer assay, performed in biological duplicate with three technical measurements each (*n* = 6). The visual representation was collected at the 24 h time point.

## Data Availability

The data presented in this study are available on request from the corresponding author. Authors can confirm that all relevant data are included in the article.

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
