# Peer review of "Polymyxin B Peptide Hydrogel Coating: A Novel Approach to Prevent Ventilator-Associated Pneumonia"

_ijms, 2024, doi:10.3390/ijms251910269_

Round 1

Reviewer 1 Report

Comments and Suggestions for Authors

The manuscript prepared by Wouters et al. investigates the potential efficacy of Polymyxin B peptide hydrogel-coated tubes in combating Ventilator-Associated Pneumonia. The topic is of significant importance and the manuscript is generally well-written. However, there are a few minor issues that need to be addressed before publication in IJMS:

1.          Detail the preparation steps for the antibacterial hydrogel coating for the readers, including the rationale for using acrylic acid (AA) and sulfobetaine methacrylate (DMAPS) in the hydrogel formulation.

2.          Clarify whether the Polymyxin B peptide was synthesized in-house or sourced from a supplier. Additionally, provide information on its purity, and include HPLC and mass spectrometry data for the peptide.

3.          Specify which residue of the Polymyxin B peptide is covalently bonded to the surface of the hydrogel-coated PVC tube when using EDC and NHS for modification.

4.          Discuss why Polymyxin B might still be released from the tube even though it is chemically modified onto the surface.

5.          Estimate the cost of preparing Polymyxin B peptide-coated PVC tubes to aid in future clinical applications.

6.          Examine the potential antibacterial mechanisms of the Polymyxin B peptide-coated PVC tubes.

Reviewer 2 Report

Comments and Suggestions for Authors

This manuscript evaluated hydrogel coating of PMB for potential application in VAP. The study design is adequate and experiments appear to support the hypothesis. However, authors should address following points for potential acceptance in the journal. 

1. Cytotoxicity studies: It is unclear how much (concentration of PMB) materials were used. PMB is known to be cytotoxic (Future Microbiol. 2013 (6):711-24. doi: 10.2217/fmb.13.39, Int J Mol Sci. 2022 23(9):4558. doi: 10.3390/ijms23094558.). These articles should be mentioned. 

2. IR studies: It is not clear differences between coated and coated with PMB materials. Can they predict conformations of PMB? IR of free PMB should be shown. 

3. Raman studies: Spectra should be labelled. It is not clear how PMB affecting Raman spectra? Can authors use difference spectra (IR and Raman) to show PMB effect. 

4. PMB release assay: Authors should provide appropriate references for the assay method used in the study. 

5. Antibacterial evaluation: Biofilm assays and killing of planktonic bacteria should be different. Authors should clarify this. 

Round 2

Reviewer 2 Report

Comments and Suggestions for Authors

No more comments.